# From Awareness to Action: Addressing Knowledge Barriers and Promoting Herpes Zoster Vaccination in Chinese Rheumatic Disease Patients

**DOI:** 10.3390/vaccines13070674

**Published:** 2025-06-24

**Authors:** Yan Geng, Bofan Lu, Wenhui Xie, Yu Wang, Xuerong Deng, Juan Zhao, Guangtao Li, Xiaohui Zhang, Zhibo Song, Zhuoli Zhang

**Affiliations:** Department of Rheumatology and Clinical Immunology, Peking University First Hospital, No. 8, Xishiku Street, West District, Beijing 100034, China; gengyan8487@163.com (Y.G.); bofanlu011210@163.com (B.L.); xiewh@pku.edu.cn (W.X.); 13693374001@163.com (Y.W.); 06485@pkufh.com (X.D.); juanzi810819@163.com (J.Z.); leegt@163.com (G.L.); beiyizhangxiaohui@163.com (X.Z.); iamsongzhibo@126.com (Z.S.)

**Keywords:** survey, herpes zoster, vaccines, rheumatic disease, knowledge, attitudes

## Abstract

**Aim:** To investigate the knowledge of and attitudes towards herpes zoster (HZ) and its vaccination, as well as the vaccination status of Chinese patients with rheumatic disease. **Method:** A face-to-face questionnaire survey was conducted among patients visiting the Department of Rheumatology and Clinical Immunology, Peking University First Hospital from 1 March to 30 April 2024. Information on HZ infection and vaccination status was recorded. The questionnaire assessed their knowledge of HZ and the HZ vaccine with nine questions, scoring one point for each correct answer, resulting in a total score ranging from zero to nine. Attitudes toward HZ and vaccines were measured by a five-point Likert scale, with scores ranging from one (“strongly disagree”) to five (“strongly agree”). Factors associated with knowledge and attitude scores were analyzed using an ordinal logistic regression. **Results:** A total of 1036 patients completed the questionnaire, with a mean age of 47.1 years, and 79.3% were females. The three most prevalent diseases were systemic lupus erythematosus (32.4%), rheumatoid arthritis (26.3%) and Sjögren’s syndrome (9.9%). A total of 243 patients (23.5%) reported a history of HZ or current HZ infection. Only 2.0% of the patients had been vaccinated, while 51.0% expressed willingness to be vaccinated in the future. The median knowledge score was four (2, 5) (ranging from zero to nine), and the median attitude score was 19 (17, 20) (ranging from 5 to 25). Factors associated with higher knowledge scores included being female (β = 0.448, *p* = 0.001), having a higher educational level (β = 0.355, *p* < 0.001), having a higher monthly income (β = 0.191, *p* = 0.008) and having comorbidities (β = 0.275, *p* = 0.023). Factors associated with higher attitude scores included being female (β = 0.279, *p* = 0.035), having a higher monthly income (β = 0.196, *p* = 0.037) and possessing a higher education level (β = 0.310, *p* = 0.045). **Conclusions:** Patients with rheumatic disease in China exhibit a low level of cognition regarding HZ as well as its vaccine, and the vaccination rate is very low. To improve the understanding and prevention awareness of HZ, health education should be intensified, particularly targeting males, those with lower levels of education and lower-income patients.

## 1. Introduction

Herpes zoster (HZ) is an infectious disease characterized by painful skin blisters caused by the varicella-zoster virus (VZV). Post-herpetic neuralgia significantly impairs one’s quality of life [1]. Epidemiological data indicate that approximately 99.5% of adults aged ≥50 years have been previously exposed to VZV, placing them at persistent risk for viral reactivation [2]. This risk is notably heightened among individuals with rheumatic and immunological disorders, who often exhibit impaired immune surveillance due to disease pathophysiology or immunosuppressive therapies. In these populations, VZV reactivation not only contributes to increased morbidity but also imposes substantial therapeutic and quality-of-life burdens [3]. Given the widespread prevalence of latent VZV and the heightened vulnerability of immunocompromised hosts, a comprehensive understanding of herpes zoster risk and prevention strategies in these groups is of paramount importance.

The global incidence rate of HZ in the general population is estimated to be 3–5 per 1000 person-years, and it increases with age. In China, the incidence among individuals aged 50 years and older is 2.8–5.8 per 1000 person-years, with a lifetime risk of developing the disease exceeding 30% [4,5]. Patients with rheumatic diseases are more susceptible to HZ. Several factors contribute to this heightened vulnerability, including the use of glucocorticoids (GCs) and immunosuppressants, advancing age and an elevated neutrophil-to-lymphocyte ratio [6]. The incidence of HZ is significantly higher in patients with rheumatic diseases compared to the general population: 2.3 times higher in rheumatoid arthritis (RA), 4.0 times higher in systemic lupus erythematosus (SLE), 3.9 times higher in inflammatory myopathies, and 1.7 times higher in primary Sjögren’s syndrome (pSS) [7]. Furthermore, patients with RA and SLE not only exhibit a higher initial risk of HZ but also face increased recurrence. Recurrence rates in these populations, when compared to individuals without immunocompromising conditions, are markedly elevated at 2.43 and 2.75 per 100 person-years for RA and SLE, respectively [8].

HZ is associated with a range of complications that can significantly impair patient health and quality of life. Notably, neurologic involvement may result in postherpetic neuralgia (PHN), a chronic pain condition that can persist for months or even years following the acute phase of infection. Ophthalmic HZ may lead to serious corneal damage and vision loss. HZ otitis externa may result in auditory impairment and facial nerve palsy. The severity and potential irreversibility of these complications highlight the critical importance of timely and effective preventive strategies, particularly vaccination, in reducing the burden of disease [9]. Patients with SLE and RA are at significantly increased risk of developing PHN following HZ compared to individuals without autoimmune conditions. Specifically, the risk of PHN is 1.8-fold higher in patients with SLE and 1.1-fold higher in those with RA. Beyond pain-related complications, HZ in patients with RA is associated with broader systemic risks: RA patients who develop HZ experience a 27% increased risk of stroke and an 18% increase in all-cause mortality compared to RA patients without HZ [10,11]. The co-occurrence of herpes zoster (HZ) significantly increases the medical burden in patients with rheumatic diseases by raising healthcare use, prolonging treatment and elevating complication risks. Evidence indicates that individuals with RA who develop HZ face a significantly elevated risk of hospitalization compared to RA patients without HZ involvement [12]. Among patients diagnosed with HZ, 6.6% of hospital admissions are directly attributable to the HZ episode itself, while an additional 28.6% are linked to comorbidities or underlying conditions [13,14]. These findings highlight not only the increased incidence of HZ in immunocompromised populations—particularly those with rheumatic diseases—but also its role in escalating the demand for inpatient care. Managing both conditions poses clinical challenges, emphasizing the need for proactive prevention—especially vaccination—and close monitoring to reduce morbidity and healthcare costs [15]. In addition, previous studies have demonstrated that oncological patients—particularly those receiving chemotherapy—are at a significantly elevated risk for HZ, with increased rates of hospitalization and related complications. As such, this population represents a high-risk group that must be carefully considered when evaluating the impact of HZ on the healthcare burden [16].

HZ can impact the treatment and prognosis of rheumatic diseases. Administering the HZ vaccine can reduce the risk of HZ infection in rheumatic patients over 50 years old by 51–70%. According to the 2019 European League Against Rheumatism (EULAR) recommendations for vaccination in adult patients with autoimmune inflammatory rheumatic diseases and the 2023 Chinese expert consensus, high-risk individuals are advised to prioritize vaccination [17,18]. The high-risk group includes individuals over the age of 50 and patients on immunosuppressive medications, including GC, immunosuppressants and Janus kinase (JAK) inhibitors. A population-based study evaluated the risk of HZ in patients with RA receiving JAK inhibitors compared to those treated with conventional synthetic disease-modifying antirheumatic drugs (csDMARDs). The findings demonstrated a significantly elevated risk of HZ among patients treated with JAK inhibitors—such as tofacitinib, baricitinib and upadacitinib—relative to those receiving csDMARD therapy, underscoring the importance of targeted preventive measures in this population [19]. In China, generic JAK inhibitors, which are available at very low prices, are widely used, leading to a rise in HZ infection cases. The recombinant zoster vaccine is considered to have superior efficacy and safety profiles, especially for individuals at higher risk or with compromised immune systems [20,21,22]. Currently, two types of HZ vaccines are available in China: the recombinant zoster vaccine (Shingrix, GlaxoSmithKline) and the live attenuated vaccine (Ganwei, Changchun Baike Biotechnology), introduced in 2020 and 2023, respectively. In addition to these licensed vaccines, multiple live-attenuated and recombinant subunit vaccines are undergoing clinical evaluation, while DNA- and mRNA-based candidates remain in preclinical stages of development. The recombinant zoster vaccine, although clinically recommended, has not yet been incorporated into China’s national immunization program. It requires a two-dose regimen administered 2–6 months apart, typically delivered at community health centers, with the full cost borne by individuals. In alignment with the 2023 Chinese expert consensus, HZ vaccination is recommended for patients with autoimmune rheumatic diseases during periods of disease stabilization. Nevertheless, a recent national survey reported that the overall vaccination rate among individuals aged 40 years and older remains exceedingly low, at only 0.79%, primarily due to limited public awareness and inadequate vaccine supply [23,24].

It is currently unclear whether expert consensus influences the knowledge and behavior of Chinese patients with rheumatic diseases. The objective of the study is to investigate the knowledge of and attitudes towards HZ and its vaccine, as well as the vaccination status of Chinese patients with rheumatic disease.

## 2. Patients and Methods

### 2.1. Study Design

A questionnaire survey was administered via face-to-face interviews at the Department of Rheumatology and Clinical Immunology, Peking University First Hospital from 1 March to 30 April 2024. A structured electronic questionnaire was used to gather information from patients. Inclusion criteria were as follows: (1) patients aged 18 years or older, (2) with confirmed diagnosis of a rheumatic disease by a rheumatology specialist, and (3) with the ability to understand and complete the questionnaire. Exclusion criteria included the following: (1) patients without a confirmed rheumatic diagnosis, (2) those under 18 years of age, and (3) individuals with cognitive or communication impairments that could interfere with questionnaire completion.

To ensure adequate representativeness of the study population, the required sample size was estimated using a margin of error (E) of 0.05 and a 95% confidence level (Z = 1.96). To account for the potential clustering effects and the non-random nature of convenience sampling, a design effect (DE) of 2 was applied. Additionally, a 10% allowance for potential sample loss was incorporated into the calculation. Based on these parameters, the final estimated sample size was 856 participants. A total of 1036 valid questionnaires were ultimately collected, exceeding the estimated requirement and thereby ensuring adequate statistical power and representativeness of the study findings.

The study was approved by Institutional Review Board of Peking University First Hospital. Participants were fully informed about the purpose of the study. It was also clearly stated that participation was entirely voluntary, and participants could withdraw at any time. They were assured that their data would remain confidential, with no personal details being disclosed.

### 2.2. Questionnaire Design

The survey is divided into four sections. Part 1 collects basic information about the patients, including demographic characteristics, such as gender, age, type of disease, level of education, and income level. Part 2 consists of 9 questions about knowledge of HZ and vaccines, covering topics such as the concept of HZ, its causes, risk factors, contagiousness, symptoms, potential for recurrence, associated complications, as well as the availability and efficacy of the HZ vaccine. Part 3 assesses their attitude through 5 questions regarding the severity of and susceptibility to HZ, as well as necessity, effectiveness and safety of the vaccine. Part 4 investigates intentions and status regarding HZ infection and vaccination. The Cronbach’s alpha reliability coefficients for the cognitive, attitudinal and vaccination status sections of the questionnaire were 0.81, 0.77 and 0.81, respectively.

The questionnaire was initially designed by two investigators (YG and ZZ). Based on the pilot testing results from the sample population, the questionnaire was then modified and refined by an expert panel of healthcare professionals specializing in epidemiology and biostatistics at Peking University to align with the objectives of the study.

### 2.3. Structure Questionnaire (English Version)

#### 2.3.1. Basic Information

(1)Name:(2)Gender: Male; Female(3)Age (years):(4)Educational Level: Junior high school or lower; High school or vocational secondary education; Associate degree or undergraduate; Master’s degree or higher(5)Monthly Income (CNY): ≤2000; 2000–5000; 5000–10,000; ≥10,000(6)Type(s) of autoimmune diseases:

Rheumatoid arthritis; Systemic lupus erythematosus; Sjögren’s syndrome; Systemic sclerosis; Inflammatory myopathy; Spondyloarthritis; Antiphospholipid syndrome; Vasculitis; Adult-onset Still’s disease; Osteoarthritis; Relapsing polychondritis; Gout and hyperuricemia; IgG4-related disease; Fibromyalgia; Autoimmune liver disease; Osteoporosis; Other

(7)Other chronic comorbidities (multiple choices):

Chronic obstructive pulmonary disease; Hyperlipidemia; Hypertension; Diabetes; Cardiovascular disease; Cerebrovascular disease; Chronic kidney disease; Other

#### 2.3.2. Knowledge of Herpes Zoster and Its Vaccine

(8)Is herpes zoster primarily classified as a skin disease?

Yes; No; Not sure

(9)Is herpes zoster caused by a virus?

Yes; No; Not sure

(10)Which of the following factors are associated with an increased risk of developing herpes zoster? (multiple choices)

Older age; Male; Female; History of chickenpox; Immunocompromised individuals; Fatigue and stress

(11)Which of the following are common clinical manifestations and symptoms of herpes zoster? (multiple choices)

Papules; Blisters; Pain; Blindness; Hearing loss

(12)Can individuals be infected with herpes zoster through contact with a patient who has herpes zoster?

Yes; No; Not sure

(13)Can herpes zoster recur following recovery?

Yes; No; Not sure

(14)Which of the following are complications of herpes zoster? (multiple choices)

Neuralgia; Myocarditis; Ocular complications; Meningitis; Arthritis

(15)Is there a vaccine available for the prevention of herpes zoster?

Yes; No; Not sure

(16)Is the herpes zoster vaccine effective in preventing herpes zoster?

Yes; No; Not sure

#### 2.3.3. Attitudes Towards Herpes Zoster and Its Vaccine

(17)Herpes zoster is a disease characterized by severe pain and complications.

Strongly agree; Agree; Neutral; Disagree; Strongly disagree

(18)The risk of developing herpes zoster is elevated in middle-aged and elderly individuals, particularly those over the age of 50.

Strongly agree; Agree; Neutral; Disagree; Strongly disagree

(19)Patients with rheumatic and autoimmune diseases, particularly those over 50 years of age or receiving immunosuppressive medications, should be considered for vaccination against herpes zoster.

Strongly agree; Agree; Neutral; Disagree; Strongly disagree

(20)What is the efficacy of the herpes zoster vaccine in preventing the onset of herpes zoster?

Very effective; Quite effective; Neutral; Not very effective; Not effective at all

(21)What is the safety profile of the herpes zoster vaccine?

Very safe; Quite safe; Neutral; Not very safe; Very unsafe

#### 2.3.4. Intentions and Practices

(22)Have you ever had or do you currently have herpes zoster?

Yes; No; Not sure

(23)Do you think there is a vaccine available to prevent herpes zoster?

Yes; No; Not sure

(24)Have you received the herpes zoster vaccine?

Yes; No; Not sure

(25)What are the main factors that influence your willingness to receive the herpes zoster vaccine? (multiple choices)

Concerns about vaccine safety; Considerations regarding vaccine efficacy; Number of required doses; Uncertainty about vaccination locations; High cost of vaccine; Other

(26)Would you like to learn more about herpes zoster and its vaccine?

Yes; No; Not sure

(27)How did you obtain information about herpes zoster and its vaccines? (multiple choices)

Through the television; Through the internet; General hospitals; Community health service centers; Family or friends; Other

(28)Have you discussed or shared information regarding herpes zoster and its vaccination with your family members or close friends?

Yes; No; Not sure

(29)Would you consider receiving the herpes zoster vaccine for yourself?

Yes; No; Not sure

(30)Would you be willing to accept the total cost of the herpes zoster vaccine within the range of 2000–3000 CNY?

Yes; No; Not sure

(31)Have you ever received a self-paid vaccine?

Yes; No; Not sure

(32)What type of self-paid vaccine have you received? (multiple choices)

Influenza vaccine; Pneumonia vaccine; Hepatitis vaccine; Herpes zoster vaccine; Other

### 2.4. Data Analysis

Data analysis was conducted using SPSS version 26.0. The questionnaire consisted of 9 items designed to assess participants’ knowledge about HZ and vaccines. Each correct response was awarded 1 point, resulting in a total possible score ranging from 0 to 9 points. Attitudes towards HZ and vaccines were measured using a 5-point Likert scale, with responses ranging from “strongly disagree” (1 point) to “strongly agree” (5 points). Descriptive statistical methods were employed to evaluate the overall demographic profiles, levels of knowledge and attitudinal tendencies of the patients. Ordinal logistic regression analysis was used to identify factors associated with knowledge of and attitudes towards HZ.

## 3. Results

### 3.1. Demographic Characteristics of the Participants

A total of 1036 patients participated in the survey and completed the questionnaire. Of these, 822 (79.3%) were female, with an average age of 47.1 ± 15.3 years. The educational backgrounds of the participants varied significantly: 28.0% had attended junior high school or had lower levels of education, 21.0% had completed high school or vocational secondary education, 43.5% had attained junior college or undergraduate degrees and 7.5% had progressed to graduate-level studies or higher. The patients’ monthly income levels are fairly evenly distributed: 32.4% earn between 2000 and 5000 CNY per month, while 30.2% earn between 5000 and 10,000 CNY. Those with monthly incomes below 2000 CNY and above 10,000 CNY account for 20.8% and 16.6% of the patients, respectively (Figure 1, Table 1).

Among the patients enrolled in our study, the three most prevalent rheumatic diseases were SLE with 336 patients (32.4%), RA with 272 patients (26.3%) and pSS with 102 patients (9.9%). Other rheumatic diseases included systemic sclerosis, spondyloarthropathy, antiphospholipid syndrome, vasculitis, adult-onset Still’s disease and others. A total of 364 patients had comorbidities, with 20.8% having hypertension, 8.8% having hyperlipidemia and 4.9% having diabetes (Figure 1, Table 1).

### 3.2. Knowledge of, Attitude Towards, and Factors Associated with HZ and HZ Vaccine

The median knowledge score was four (2, 5), with scores ranging from zero to nine, indicating substantial variability in awareness levels. Notably, 8.5% of participants scored zero, while only 0.2% achieved the maximum score. The majority of respondents scored between one and eight, with the most frequent score being five (17.4%). Scores of one, two, three, four, five, six, seven and eight were reported by 11.0%, 13.3%, 15.7%, 16.6%, 11.5%, 5.0% and 0.8% of patients, respectively. These findings suggest a moderate level of knowledge in the patient population, with a central tendency toward mid-range scores. Approximately 53.4% of the patients recognized that HZ is characterized by skin involvement, and 54.5% were aware that the disease is caused by a virus. Additionally, 57.9% of patients knew that HZ could lead to the development of neuralgia. Furthermore, 74.2% of the patients were aware of the risk factors for HZ, such as fatigue, stress and an advanced age. However, only 45.0% were aware of the existence of a vaccine against HZ.

In contrast, the median score for attitude was 19 (17, 20), with scores ranging from 5 to 25. The median attitude scores for effectiveness, safety, necessity of the HZ vaccine, susceptibility to HZ infection and severity of HZ infection were three, two, two, two, and two, respectively (on a scale of one to five).

An ordinal logistic regression analysis was conducted to identify factors associated with knowledge of and attitude towards HZ and HZ vaccination. As shown in Table 2, the results demonstrated a positive correlation between being female (β = 0.448, *p* = 0.001), having a higher monthly income (β = 0.191, *p* = 0.008), having a higher education level (β = 0.355, *p* < 0.001) and having comorbidities (β = 0.275, *p* = 0.023) and the participants’ knowledge score. Similarly, being female (β = 0.279, *p* = 0.035), having a higher monthly income (β = 0.196, *p* = 0.037) and possessing a higher education level (β = 0.310, *p* = 0.045) are positively correlated with a higher attitude score (Table 3).

The ordinal logistic regression models demonstrated an acceptable fit to the data. For the knowledge scale, the model yielded a pseudo-R^2^ value of 0.25, with the non-significant Pearson chi-square goodness-of-fit test (*p* = 0.781) indicating an adequate model fit. Similarly, for the attitude scale, the pseudo-R^2^ was 0.27, and the Pearson chi-square test was also non-significant (*p* = 0.847), further supporting the model’s adequacy. The proportional odds assumption was met for both models, suggesting that the use of an ordinal logistic regression was appropriate, and the models are robust.

The assessment of multicollinearity among independent variables showed low variance inflation factor (VIF) values: gender (1.082), age (1.097), education level (1.735), monthly income (1.742) and comorbidities (1.115). All VIF values were close to one, indicating minimal multicollinearity and weak correlations among predictors. These findings confirm the stability and reliability of the regression models.

### 3.3. Herpes Zoster Vaccination Intention and Behavioral Patterns

Among the surveyed participants, 243 (23.5%) reported a past or present diagnosis of HZ. Additionally, 184 patients (17.8%) were aware that HZ vaccine is available to prevent HZ infection; however, only 21 patients (2.0%) had received an HZ vaccination. The primary factors influencing patients’ willingness to receive the HZ vaccination include concerns about the safety of the vaccine (60.1%), concerns about the efficacy of the vaccine (53.3%), the number of required vaccinations (53.3%), a lack of knowledge of vaccination sites (51.8%) and the high cost of the vaccine (41.8%) (Figure 2).

A total of 486 (46.9%) patients expressed a desire to learn more about HZ and its vaccination. A significant portion of the patients, 56.1% and 15.6%, respectively, indicated that they had attained information regarding HZ and HZ vaccinations through the internet and television channels. A total of 21.2% and 17.2% of patients reported acquiring knowledge from general hospitals and community health service centers, respectively. A total of 16.1% of the patients acquired relevant information from family members and friends (Figure 3). Encouragingly, 82.1% of patients expressed a willingness to proactively share their knowledge with their family or friends. Furthermore, 51.0% of patients indicated a predisposition towards receiving the HZ vaccine in the future.

As for the vaccine costs, 33.1% of patients stated that they would accept the total cost of the HZ vaccine to be within the range of 2000–3000 CNY. Among the surveyed individuals, 28.9% reported having received self-paid vaccinations. The influenza vaccine had the highest uptake, representing 57.9% of self-paid vaccinations administered. This was followed by the hepatitis and pneumonia vaccine, which accounted for 28.4% and 16.7%, respectively.

## 4. Discussion

Herpes zoster (HZ) is a viral infection that can cause severe neuralgia and complications. Patients with rheumatic diseases are at a higher risk of contracting HZ compared to the general population. Guidelines recommend that high-risk individuals, including those with rheumatic diseases, receive the HZ vaccine. Accumulating evidence indicates that HZ vaccination not only significantly reduces the incidence of HZ but also contributes to a meaningful decline in HZ-related hospital admissions. These findings underscore the substantial public health value of vaccination as a preventive strategy, particularly in mitigating the healthcare burden associated with HZ. Promoting vaccine uptake among high-risk populations may play a critical role in decreasing hospitalization rates and improving overall clinical outcomes [25]. However, there are limited real-world data on the knowledge of, attitudes towards, and practices related to HZ and its vaccination among Chinese patients with rheumatic diseases. This study aims to fill this gap.

Vaccine literacy, encompassing the ability to obtain, comprehend, and utilize vaccine-related information, plays a pivotal role in facilitating informed health decisions. Our findings reveal substantial knowledge deficits concerning HZ and its prevention, highlighting an urgent need to strengthen vaccine literacy among the general population. Addressing these gaps through transparent communication, targeted health education initiatives and proactive healthcare provider–patient interactions is critical to enhancing vaccine acceptance and uptake. Notably, responses to survey items exposed significant misunderstandings about HZ disease and available preventive measures, particularly within vulnerable groups. These insights underscore the importance of integrating vaccine literacy strategies into public health programs to optimize vaccination coverage and reduce disease burden.

Our study revealed that patients had low knowledge and attitude scores regarding HZ and its vaccine. Notably, female participants, those with higher levels of education, and individuals with higher monthly incomes exhibited relatively higher knowledge and attitude scores. These findings are consistent with surveys from urban residents in China [26,27,28,29]. Gender differences in health knowledge may be influenced by various sociodemographic and cultural factors. Females tend to be more proactive in seeking health information and often serve as primary caregivers in families, which may be associated with a higher awareness of HZ.

Additionally, higher-income individuals may have greater access to healthcare services and information, may be associated with better knowledge of and attitude towards preventable diseases and vaccinations. Our findings also align with surveys conducted in Hong Kong and Saudi Arabia, indicating that education positively impacts HZ and vaccination knowledge and attitude scores. This underscores the importance of educational interventions to enhance the understanding and awareness of HZ [30,31,32,33]. We also found that patients with comorbidities had higher knowledge scores, which may be related to increased healthcare interactions, higher health literacy and targeted health education from healthcare providers. These findings highlight the importance of tailored health education and communication strategies for individuals with rheumatic disease to further enhance their knowledge and potentially increase vaccination uptake [34]. Further research is warranted to assess the effectiveness of targeted interventions in enhancing herpes zoster vaccination uptake.

In this survey, 23.5% of the patients reported having had HZ or were currently suffering from HZ; however, only 2% of them had been vaccinated against HZ. Despite the high incidence of HZ infection among the rheumatic disease patients in our survey, the vaccination rate remains low. The main factors influencing the uptake of the HZ vaccine include concerns about the safety and efficacy of the vaccine, the number of required doses, a lack of awareness regarding vaccination sites and the high cost of the vaccine. These factors are consistent with the findings of a 2023 survey conducted in urban residents in selected regions of China regarding HZ vaccination. The understanding of the vaccine’s efficacy and safety can be strengthened through education by healthcare professionals. Additionally, awareness of vaccination sites requires community outreach. Furthermore, adjustments in vaccine pricing may lead to an increased willingness to be vaccinated in the future [35].

In our study, 51.0% of patients expressed willingness to consider HZ vaccination in the future. This finding is slightly lower than the pooled vaccination willingness rate of 55.7% reported in a 2023 meta-analysis conducted by the World Health Organization [36], but higher than the rate of 35.1% in the Chinese general population [29]. This phenomenon may be attributed to the limited knowledge of and prevalent misconceptions surrounding the HZ vaccine. Notably, some individuals perceive HZ as an uncommon disease, while others harbor concerns regarding the vaccine’s efficacy and safety, which may contribute to vaccine hesitancy. These highlight the need for targeted educational approaches and incentives to enhance vaccination uptake in this vulnerable population.

Nearly half of the respondents expressed a desire to learn more about HZ and vaccination in the future. Preferred learning channels included the internet, healthcare institutions, and family and friends. A similar study showed that patients were more likely to get the vaccination if recommended by a healthcare professional [32]. Consistent with this, the present study found that physician endorsement from practitioners in general hospitals and community health service centers significantly influenced participants’ intention to vaccinate. This observation is supported by the Health Belief Model (HBM), which posits that vaccination behavior is modulated by factors such as perceived barriers, self-efficacy and external cues to action—including guidance from healthcare providers. Within the Chinese healthcare context, physicians are generally regarded as authoritative figures due to their clinical expertise and institutional trust. Consequently, their recommendations are often perceived as credible and strongly influence health-related behaviors, including decisions regarding immunization [37].

Leveraging diverse channels for educational outreach could be an effective strategy to address knowledge gaps and promote informed decision making regarding HZ vaccination. To strengthen public health interventions, enhanced economic support is essential. Government and health insurance agencies could consider implementing vaccine subsidies to reduce the out-of-pocket burden on individuals, particularly among vulnerable populations. Concurrently, optimizing healthcare delivery is critical. This includes improving training programs for healthcare providers to ensure they possess sufficient knowledge and confidence to recommend HZ vaccination effectively. Intersectoral collaboration among health commissions, insurance departments, community organizations and other relevant stakeholders is vital for the coordinated promotion of vaccination programs. Physician engagement also plays a pivotal role. Integrating vaccine education into routine clinical care for patients with rheumatic diseases can facilitate personalized discussions on vaccine safety and timing. Furthermore, collaboration between community health centers and rheumatology and immunology departments of tertiary care hospitals in vaccination efforts can strengthen referral pathways and increase vaccination rates.

We are aware of some limitations in the study. Firstly, data collection relied on participants’ self-reports, although the involvement of certified rheumatologists as investigators helped ensure the validity and reliability of the findings. This approach notably reduced the risks of under-reporting and recall bias. Another limitation is its single-center design, which may affect the generalizability of the findings. To mitigate potential biases, we recruited a sample size appropriate to the study objectives and statistical requirements and achieved a high response rate. Rigorous quality control measures were implemented throughout the data collection process, including the development of detailed protocols and standardized operating procedures (SOPs) to ensure consistency across all stages of the study. Furthermore, all data were collected by trained specialists in rheumatology and clinical immunology, thereby enhancing the reliability and internal validity of the results. Future research should aim to expand to multiple centers to enhance the generalizability of the results and provide a more comprehensive understanding of and better attitudes towards HZ vaccination among diverse populations.

In addition, the most frequently reported rheumatic diseases in our study population were SLE (32.4%), RA (26.3%) and SS (9.9%). While we recognize the importance of disease-specific analyses, the limited sample sizes within many rheumatic disease subtypes precluded meaningful subgroup comparisons due to the reduced statistical power and the potential for unreliable estimates. Consequently, we prioritized an integrated analysis of the entire cohort to generate a generalizable understanding of vaccination behaviors among patients with rheumatic diseases. We acknowledge the value of more granular, subtype-specific investigations and recommend these as a priority for future research with adequately powered samples. Last but not least, the overall HZ vaccination rate was notably low, resulting in a substantial imbalance between vaccinated and unvaccinated individuals. This asymmetry compromised the statistical power for inferential analyses and introduced the potential for estimation instability and bias. Thus, we adopted a descriptive and exploratory analytical approach to preserve the validity of our findings. Nonetheless, we remain committed to conducting more robust multivariate analyses in future studies as larger and more balanced datasets become available.

## 5. Conclusions

Patients with rheumatic disease exhibit a poor understanding of HZ and its vaccination, leading to having little motivation to get the vaccination and consequently a low vaccination rate. More attention should be given to male patients, individuals with lower educational levels and those with low incomes to enhance their understanding of HZ and vaccines. It is essential to implement health education through various channels and formats, involving both societal and healthcare institutions. Enhancing the awareness and understanding of prevention approaches is fundamental for reducing the risk of HZ infection among patients with rheumatic disease.

## Figures and Tables

**Figure 1 vaccines-13-00674-f001:**
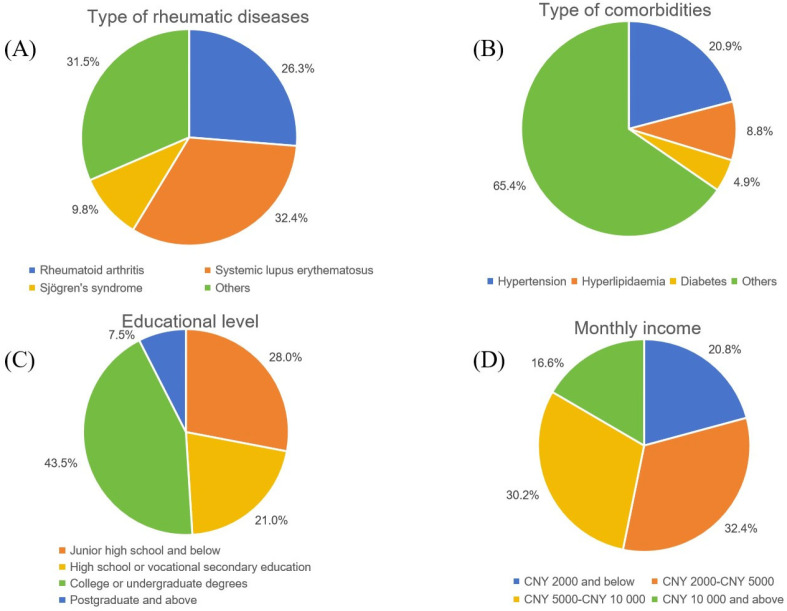
Sociodemographic and general characteristics of the participants: (**A**) type of rheumatic diseases (including SLE, RA, pSS and other rheumatic diseases); (**B**) type of comorbidities (with hypertension, hyperlipidemia, diabetes and other comorbidities); (**C**) educational level (junior high school and below, high school or vocational secondary education, college or undergraduate degrees, and postgraduate and above); (**D**) monthly income (including 2000 CNY and below, 2000–5000 CNY, 5000–10,000 CNY and 10,000 CNY and above). CNY, China Yuan.

**Figure 2 vaccines-13-00674-f002:**
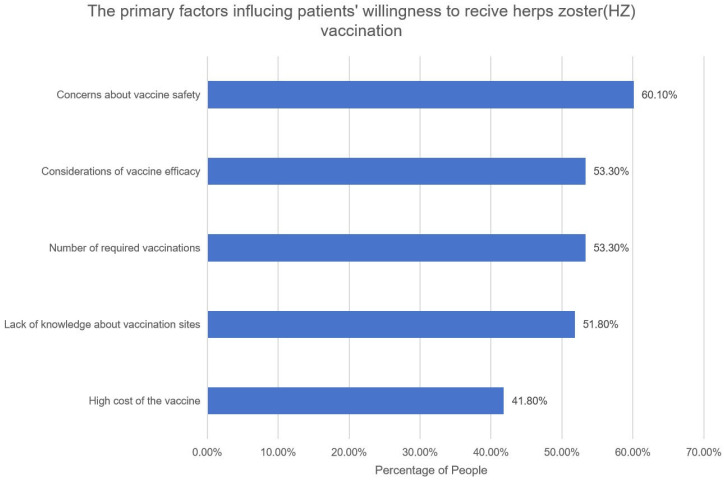
The primary factors influencing patients’ willingness to receive herpes zoster (HZ) vaccination. Bar graph depicting the primary reasons influencing patients’ willingness to receive the HZ vaccine. Each bar represents a distinct reason, and the corresponding percentage reflects the proportion of participants who gave that reason.

**Figure 3 vaccines-13-00674-f003:**
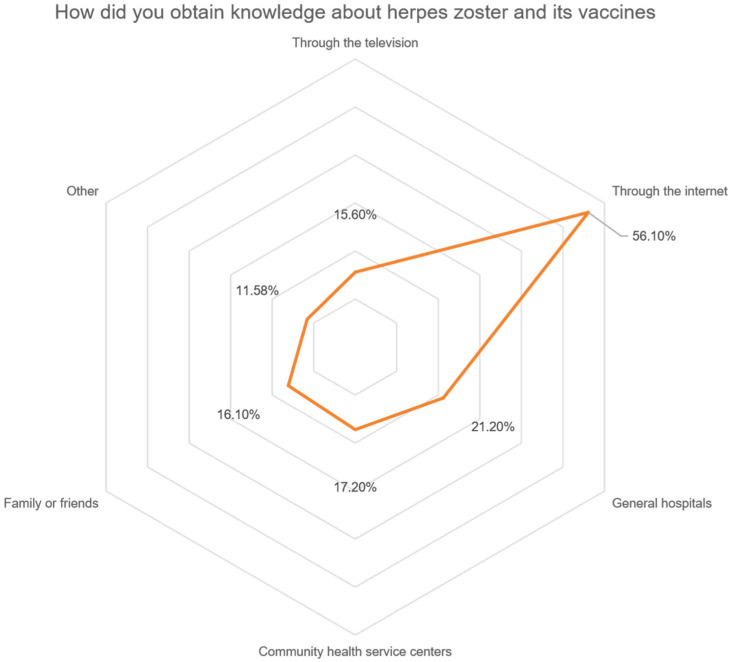
How did participants’ attain information on herpes zoster and its vaccines? Radar chart illustrating the distribution of knowledge acquisition channels among participants. Each axis represents a specific channel, and the corresponding value indicates the percentage of patients who reported obtaining HZ-related information through that channel.

**Table 1 vaccines-13-00674-t001:** Sociodemographic and general characteristics of the participants.

Demographic Data	Number (%)
Age (years)	47.1 ± 15.3
Gender (female)	822 (79.3)
Education level	
junior high school or lower	290 (28.0)
high school or vocational secondary education	217 (21.0)
junior college or undergraduate degrees	451 (43.5)
graduate-level studies or higher	78 (7.5)
Monthly income level	
≤2000 CNY	215 (20.8)
2000–5000 CNY	336 (32.4)
5000–10,000 CNY	317 (30.2)
>10,000 CNY	372 (16.6)
Diagnosis	
Systemic lupus erythematosus	336 (32.4)
Rheumatoid arthritis	272 (26.3)
Sjogren’s syndrome	102 (9.9)
Systemic sclerosis	11 (1.1)
Spondyloarthritis	59 (5.7)
Antiphospholipid syndrome	26 (2.5)
Vasculitis	41 (4.0)
Adult-onset Still’s disease	13 (1.3)
Other CTDs	176 (17.0)
Comorbidities	
Hypertension	216 (20.8)
Hyperlipidemia	91 (8.8)
Diabetes	51 (4.9)
Cardiovascular Disease	48 (4.6)
Chronic obstructive pulmonary disease	10 (1.0)
Chronic kidney disease	38 (3.7)
Other comorbidities	44 (4.2)

CNY, China Yuan.

**Table 2 vaccines-13-00674-t002:** Factors associated with knowledge score of HZ and HZ vaccination (ordinal logistic regression).

	β-Value (95%CI)	*p*-Values
Gender (ref. male)	0.448 (0.713, 0.723)	0.001
Age (years)	0.003 (−0.002, 0.009)	0.201
Monthly income (ref. ≤ 2000 CNY)	0.191 (0.049, 0.333)	0.008
Education level (ref. primary and below)	0.355 (0.208, 0.503)	<0.001
Comorbidity (ref. 0)	0.275 (0.039, 0.512)	0.023

ref.: reference; CNY, Chinese Yuan.

**Table 3 vaccines-13-00674-t003:** Factors associated with attitude score towards HZ and HZ vaccination (ordinal logistic regression).

	β-Value (95% CI)	*p*-Values
Gender (ref. male)	0.279 (0.005, 0.553)	0.035
Monthly income (ref. ≤ 2000 CNY)	0.196 (0.054, 0.337)	0.037
Education level (ref. primary and below)	0.310 (0.163, 0.457)	0.045
Comorbidity (ref. 0)	0.151 (−0.085, 0.387)	0.387

## Data Availability

The original contributions presented in this study are included in the article. Further inquiries can be directed to the corresponding author.

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
