# Peer review of "From Awareness to Action: Addressing Knowledge Barriers and Promoting Herpes Zoster Vaccination in Chinese Rheumatic Disease Patients"

_vaccines, 2025, doi:10.3390/vaccines13070674_

Round 1
Reviewer 1 Report
Comments and Suggestions for Authors
I was invited to revise the paper entitled "From Awareness to Action: Addressing Knowledge Barriers and Promoting Herpes Zoster Vaccination in Chinese Rheumatic Disease Patients". It was a cross-sectional study aimed to evaluate attitudes towards HZ and related factors among Chinese Rheumatic Patients.
The topic is interesting and faced an important topic for public health. In my knowledge, poor studies were published from China on this topic.
Despit that, I observed several limitations:
- In introduction section Authors should report the possible impact of HZ on hospitalizations;
- Authors did not reported vaccine availability and vaccination schedule proposed in China;
- Authors should report baseline characteristics of included patients;
- Sample size estimation was totally lacking;
- Likert scale variables are discrete variables and non-normally distributed. So they can't be summarized as mean and sd and they can't be analyzed with linear regression.
Author Response
Comment 1: In introduction section Authors should report the possible impact of HZ on hospitalizations.
Response 1: We appreciate the reviewer’s insightful suggestion. In response, we have revised the Introduction to include a discussion on the impact of herpes zoster (HZ) on hospitalizations (Line73-80). Specifically, we have incorporated recent epidemiological evidence highlighting the burden of HZ-related hospital admissions, with a focus on high-risk populations such as the elderly and immunocompromised. These additions underscore the clinical and public health relevance of HZ, supporting the rationale for enhanced preventive strategies. Relevant references have been cited to substantiate these points (PMID:25201241, 34870800, 40359432).
Comment 2: Authors did not report vaccine availability and vaccination schedule proposed in China.
Response 2: We thank the reviewer for this valuable suggestion. In response, we have revised the Introduction to include a comprehensive overview of herpes zoster vaccine availability in China (Line 100-113). Specifically, we now detail the two currently licensed vaccines—Shingrix (recombinant zoster vaccine) and Ganwei (live attenuated vaccine)—along with their respective approval timelines. We have also outlined the recommended vaccination schedule in accordance with national guidelines, including the two-dose regimen for Shingrix and its administration intervals. Furthermore, we have incorporated a discussion of key challenges affecting vaccine uptake in China, such as limited public awareness and the lack of integration into the national immunization program. Relevant references have been cited to substantiate these points (PMID:40045716, 40253789).
Comment 3: Authors should report baseline characteristics of included patients.
Response 3: We appreciate the reviewer’s helpful suggestion. In response, we have added a detailed table (Table1) in the Result section presenting the baseline characteristics of the included patients. Additionally, we have included a descriptive paragraph summarizing key variables such as age, sex, disease duration, and relevant clinical parameters. These additions aim to provide a clearer and more comprehensive profile of the study population, thereby enhancing the transparency and interpretability of our findings.
Comment 4: Sample size estimation was totally lacking.
Response 4: We appreciate the reviewer’s valuable comment regarding sample size estimation (Line 129-137). In response, we have added a detailed explanation to the Methods section. To ensure adequate representativeness, the sample size was calculated using a margin of error (E) of 0.05 and a 95% confidence level (Z = 1.96). A design effect (DE) of 2 was applied to account for potential clustering and the non-random nature of convenience sampling. Additionally, we adjusted for a 10% anticipated sample loss rate, resulting in a final estimated sample size of 856 participants. Ultimately, a total of 1036 valid questionnaires were collected, exceeding the required threshold and thereby ensuring sufficient statistical power and representativeness for the study findings.
Comment 5: Likert scale variables are discrete variables and non-normally distributed. So they can't be summarized as mean and sd and they can't be analyzed with linear regression.
Response 5:
Thanks for your kind reminder. In response, we have revised the Statistical Analysis section to appropriately handle Likert scale data. These variables are now summarized using medians and interquartile ranges, rather than means and standard deviations. Furthermore, we have replaced linear regression with ordinal logistic regression for the analysis of Likert-scale outcomes, in accordance with their ordinal and non-parametric nature. Any previous instances of inappropriate use of linear regression for these variables have been removed to ensure methodological rigor. As suggested by the reviewers, changes have been made in the corresponding sections on methods (Line 252) and results part (Table2 and Table3).
We sincerely thank you for your consideration and support. We believe that the revisions have significantly improved the manuscript and addressed the reviewers’ concerns. We look forward to your favorable response.
Reviewer 2 Report
Comments and Suggestions for Authors
This study addresses a relevant and underexplored issue: awareness and uptake of herpes zoster (HZ) vaccination among Chinese patients with rheumatic diseases; the large sample size (n=1036), clear stratification by sociodemographic variables, and use of validated methods and models are commendable and support the relevance of the findings.
Here are some comments/suggestions for changes that are in my opinion needed before publication:
- The development and validation of the questionnaire require more detail. Was it piloted? Were psychometric properties (e.g., internal consistency, validity) assessed? This is crucial given the reliance on subjective measures, and must be specified.
- All patients were from a single tertiary hospital in Beijing, somewhat limiting generalizability; the authors acknowledge this but must address the implications more directly, e.g. including potential urban and socioeconomic biases. Moreover, were all possible ways to reduce the risk of bias adopted?
- While multivariate regression was performed, assumptions (e.g., multicollinearity, residual normality) are not reported in the results section. Have the authors considered sensitivity or stratified analyses? If not, why?
- The manuscript often suggests a causal link between low knowledge and low herpes zoster vaccine uptake; while this is a plausible hypothesis, the cross-sectional nature of the study never supports causal inference. Associations observed between knowledge or attitudes and vaccine behavior should be interpreted with caution. Moreover, one of the most critical determinants of adult vaccination, recommendation by a healthcare provider, seems to be insufficiently explored in both the analysis and discussion. To strengthen the interpretation, the authors might consider framing the findings within the broader construct of vaccine literacy, a multidimensional concept that encompasses not only factual knowledge but also individuals’ ability to access, understand, appraise, and apply vaccine-related information in a complex health system, since vaccine literacy also integrates community and organizational dimensions and may provide a more comprehensive framework to interpret the observed gaps in awareness and behavior among patients with rheumatic disease (1).
- (minor) There are numerous grammatical errors, typographic inconsistencies, and the figure is not adequately labelled.
- (minor) Policy implications: The discussion might benefits from a more nuanced reflection on vaccine access, reimbursement policies, and physician engagement strategies, especially given the high out-of-pocket cost in China.
1. https://pubmed.ncbi.nlm.nih.gov/37553624/
Author Response
Comment 1: The development and validation of the questionnaire require more detail. Was it piloted? Were psychometric properties (e.g., internal consistency, validity) assessed?
Response 1: We appreciate the reviewer’s insightful comment. In response, we have expanded the Methods section to provide a comprehensive account of the questionnaire development and validation process. The questionnaire was initially drafted based on a literature review and expert consultation. It was then piloted among a small sample of patients with rheumatic diseases to assess clarity, relevance, and comprehension. Based on feedback, minor revisions were made to optimize item wording and structure. To evaluate psychometric properties, internal consistency was assessed using Cronbach’s alpha, and relevant indicators of content and construct validity were also examined. These methodological details are now clearly presented in the revised manuscript (Line 152-157).
Comment 2: All patients were from a single tertiary hospital in Beijing, somewhat limiting generalizability; the authors acknowledge this but must address the implications more directly, e.g., including potential urban and socioeconomic biases.
Response 2: We thank the reviewer for this important observation. In response, we have expanded the Discussion section to more directly address the implications of sampling from a single tertiary hospital in Beijing (Line 441-449). We explicitly acknowledge the potential urban and socioeconomic biases inherent in our sample, which may limit the generalizability of our findings to other regions or healthcare settings, particularly rural or under-resourced areas. Additionally, we have discussed how these contextual factors could influence patient access to vaccination services, health literacy, and vaccine attitudes. To strengthen the internal validity of our findings, we ensured standardized questionnaire administration by trained rheumatologists using a consistent protocol. Our team also participated in regular training sessions to uphold data collection quality, and we conducted ongoing quality assurance checks to verify the accuracy and completeness of the data. Lastly, we have suggested that future multicenter studies including diverse geographic and socioeconomic populations are needed to improve external validity.
Comment 3: While multivariate regression was performed, assumptions (e.g., multicollinearity, residual normality) are not reported in the results section. Have the authors considered sensitivity or stratified analyses?
Response 3: Thanks for your kind suggestion. In response, we have revised the Statistical Analysis section to enhance the transparency and methodological rigor of our approach. First, we assessed multicollinearity among independent variables using the variance inflation factor (VIF), and no significant collinearity was detected. Given the ordinal nature of the Likert scale variables, we replaced linear regression with ordinal logistic regression, which does not require the assumption of residual normality. A brief explanation of the assumptions underlying this model has been added to the manuscript (Line 317-321).
To evaluate model fit, we report that the ordinal logistic regression models yielded Pseudo-R² values of 0.25 (knowledge scale) and 0.27 (attitude scale). The Pearson chi-square goodness-of-fit tests were non-significant (p=0.781 and p=0.847, respectively), suggesting that the models fit the data well. Additionally, the proportional odds assumption was met, supporting the robustness of the models, the sensitivity analyses may not be necessary (Line 310-316).
We note that the primary objective of our study was to examine the overall influence of a group of independent variables on vaccine-related knowledge and attitudes in the target population. As such, subgroup analysis was not a central focus of the study. Given the strength and consistency of our model results, as well as the acceptable model fit, we believe that additional stratified analyses may not be essential in this study. Nonetheless, we agree that this remains a valuable consideration for future studies aimed at exploring subgroup-specific effects.
Comment 4: The manuscript often suggests a causal link between low knowledge and low herpes zoster vaccine uptake; while this is a plausible hypothesis, the cross-sectional nature of the study never supports causal inference. Associations observed between knowledge or attitudes and vaccine behavior should be interpreted with caution.
Response 4: Thanks for your kind suggestion. We have revised the manuscript to clarify that our cross-sectional study design does not permit causal inference. We have removed or rephrased language that may have implied a direct causal relationship between knowledge levels and vaccine uptake. Instead, we now emphasize that the associations identified should be interpreted as correlational. Additionally, we have added a statement in the Discussion section explicitly acknowledging the limitations of cross-sectional studies in establishing temporality or directionality. We also highlight the need for future longitudinal or interventional studies to more rigorously assess the causal pathways linking knowledge, attitudes, and vaccination behavior.
Comment 5: The role of physician recommendation in influencing vaccine uptake is notably absent and should be discussed.
Response 5: We appreciate the reviewer’s kind suggestion. In response, we have expanded the Discussion section to address the pivotal role of physician recommendation in influencing vaccine uptake. Drawing on findings from previous studies, we highlight physician endorsement as one of the most significant determinants of adult vaccination behavior, including for herpes zoster. We have also emphasized the need for future research to investigate the impact of healthcare provider communication and engagement on vaccine acceptance (Line 412-420). Relevant references have been cited to substantiate these points (PMID:39917737).
Comment 6: There are numerous grammatical errors, typographic inconsistencies, and the figure is not adequately labelled.
Response 6: Thanks for your kind suggestion. We have conducted a thorough revision of the manuscript to correct grammatical errors and typographic inconsistencies. We have also carefully reviewed and improved the figure labeling to enhance clarity, readability, and consistency with the journal’s formatting guidelines. These revisions have been implemented throughout the manuscript to improve its overall quality and presentation.
Comment 7: Policy implications: The discussion might benefit from a more nuanced reflection on vaccine access, reimbursement policies, and physician engagement strategies, especially given the high out-of-pocket cost in China.
Response 7: We thank the reviewer for this valuable suggestion. In response, we have expanded the Discussion section to provide a more nuanced examination of the policy implications of our findings. Specifically, we now address barriers to vaccine access, including regional disparities and the financial burden associated with out-of-pocket payment for herpes zoster vaccination in China. We also discuss the potential role of reimbursement policy reforms in improving uptake and equity. Furthermore, we highlight the importance of physician engagement and recommendation as a strategic approach to enhance vaccine acceptance. These additions aim to contextualize our findings within the broader healthcare policy landscape and inform future interventions at both clinical and policy dimension (Line 424-437).
We sincerely thank you for your consideration and support. We believe that the revisions have significantly improved the manuscript and addressed the reviewers’ concerns. We look forward to your favorable response.
Reviewer 3 Report
Comments and Suggestions for Authors
This manuscript explores the knowledge, attitudes, and vaccination behavior related to herpes zoster among Chinese patients with rheumatic diseases. It is important because there are knowledge gaps and a remarkably low vaccination rate (only 2%) within a high-risk population. It provides insights that can be used for targeted health education campaigns and vaccination strategies.
The following changes should be done
- Line 298"exibited" should be corrected to "exhibited".
- Line 356 – "genernal" in "genernal characteristics" should be corrected to "general".
- Line 373 – "have reviewed and edited the output" is vague and awkward. You could writ: "have reviewed and approved the final manuscript."
- There are acronyms that have not been explained. Eg VZV, in Line 40.
- The introduction should present data on vaccine availability and uptake in China
- The introduction needs a more detailed discussion of how current guidelines are implemented in China, including potential barriers.
- The authors don’t mention if there was any validation processes, such as pilot testing of the questionaire.
- Authors should include information about the reliability of the questionnaire, eg, Cronbach's alpha.
- Additionally, there is no mention of how the sample size was determined or whether a power analysis was conducted.
- Explain and extend the information on participant recruitment procedures and inclusion or exclusion criteria.
- The regression tables should include confidence intervals
- Authors should include specific analysis, such as by specific rheumatic disease subtypes.
- The authors should explore factors influencing actual vaccination uptake using multivariate analysis.
- The role of physician recommendation in influencing vaccine uptake is notably absent and should be discussed, in the discussion because prior studies suggest this is a key factor.
- The discussion should identify a as potential weaknesses, the lack of information on provider behavior or healthcare access.
- The discussion should include more specific suggestions for public health interventions, eg integrating vaccine education into rheumatology clinic visits or partnering with community pharmacies.
- Bibliographic references should include doi when available.
Author Response
Comment 1: Line 298 "exibited" should be corrected to "exhibited".
Response 1: Thank you for your suggestions for revision. The word “exibited” has been corrected to “exhibited” in line 369 of the revised manuscript.
Comment 2: Line 356 – "genernal" in "genernal characteristics" should be corrected to "general".
Response 2: Thank you for your suggestions. The typographical error has been corrected from “genernal” to “general” in line 463 of the revised manuscript.
Comment 3: Line 373 – "have reviewed and edited the output" is vague and awkward. You could write: "have reviewed and approved the final manuscript."
Response 3: Thank you for your suggestions for revision. In accordance with the recommendation, we have revised the sentence to: “have reviewed and approved the final manuscript” in line 480 of the revised manuscript.
Comment 4: There are acronyms that have not been explained. Eg VZV, in Line 40.
Response 4: Thank you for your suggestions for revision. we have revised the manuscript to define all acronyms at their first appearance. Specifically, “VZV” has been clarified as “varicella-zoster virus” at its initial mention in line 39.
Comment 5: The introduction should present data on vaccine availability and uptake in China.
Response 5: We thank the reviewer for this valuable suggestion. In response, we have revised the Introduction to include current data on herpes zoster vaccine availability and uptake in China. This includes information on the types of vaccines approved, their introduction timeline and vaccine uptake in China. These additions provide important context for the public health relevance of our study (Line 100-113). Relevant references have been cited to substantiate these points (PMID:40045716, 40253789).
Comment 6: The introduction needs a more detailed discussion of how current guidelines are implemented in China, including potential barriers.
Response 6: We thank the reviewer for this insightful suggestion. We have expanded the Introduction to include a more detailed overview of the current Chinese guidelines on herpes zoster vaccination, including recommendations on timing and dosage. We have also analyzed potential barriers to implementation, such as limited public awareness, high out-of-pocket costs and inconsistent reimbursement policies. These additions aim to provide a clearer understanding of the real-world challenges in translating guideline recommendations into practice (Line 108-113). Relevant references have been cited to substantiate these points (PMID:40253789).
Comment 7: The authors don’t mention if there was any validation process, such as pilot testing of the questionnaire.
Response 7: Thank you for your valuable comment. We have now included a detailed description of the questionnaire development and validation process in the revised manuscript. Specifically, the questionnaire was pilot tested with a small group of patients to assess its clarity, relevance, and comprehensibility. Feedback from this pilot testing was used to refine the items and improve the overall quality of the instrument (Line 155-157).
Comment 8: Authors should include information about the reliability of the questionnaire, e.g., Cronbach's alpha.
Response 8: We appreciate the reviewer’s valuable suggestion. In response, we have updated the Methods section to include information on the reliability assessment of the questionnaire. Internal consistency was evaluated using Cronbach’s alpha, and the results indicate acceptable reliability for the main scales. In addition, we have reported relevant validity measures to support the robustness of the instrument (Line 152-154).
Comment 9: Additionally, there is no mention of how the sample size was determined or whether a power analysis was conducted.
Response 9: Thanks for your kind reminder. We have added a detailed description of the sample size estimation procedure in the Methods section. To ensure adequate representativeness, the sample size was calculated using a margin of error (E) of 0.05 and a 95% confidence level (Z = 1.96). A design effect (DE) of 2 was applied to account for potential clustering and the non-random nature of convenience sampling. Additionally, we adjusted for a 10% anticipated sample loss rate, resulting in a final estimated sample size of 856 participants. Ultimately, 1,036 valid responses were collected, exceeding the required threshold and ensuring sufficient power and representativeness for the study (Line 129-137).
Comment 10: Explain and extend the information on participant recruitment procedures and inclusion or exclusion criteria.
Response 10: We appreciate the reviewer’s helpful suggestion. In response, we have revised the Methods section to provide a more detailed description of the participant recruitment process and the inclusion and exclusion criteria. Patients were recruited from the Rheumatology and Clinical Immunology department of our hospital. Inclusion criteria were: (1) age 18 years or older, (2) confirmed diagnosis of a rheumatic disease by a rheumatology specialist, and (3) ability to understand and complete the questionnaire. Exclusion criteria included: (1) patients without a confirmed rheumatic diagnosis, (2) those under 18 years of age, and (3) individuals with cognitive or communication impairments that could interfere with questionnaire completion. These criteria were applied to ensure the relevance and reliability of the data collected (Line123-128).
Comment 11: The regression tables should include confidence intervals.
Response 11: Thank you for your suggestions for revision. We have revised the regression tables to include 95% confidence intervals for all reported odds ratios. This addition provides a clearer representation of the precision and reliability of the estimated associations (Table2 and 3).
Comment 12: Authors should include specific analysis, such as by specific rheumatic disease subtypes.
Response 12: We sincerely appreciate the reviewer’s insightful feedback regarding the inclusion of analyses by specific rheumatic disease subtypes. In our study cohort, the three most prevalent rheumatic diseases were systemic lupus erythematosus 336 (32.4%), rheumatoid arthritis 272 (26.3%), and Sjögren’s syndrome 102 (9.9%). However, given the extensive range of rheumatic diseases represented and the relatively limited sample size for many subtypes, conducting detailed subgroup analyses presents considerable challenges. The small numbers within less common disease categories limit statistical power and reduce the reliability of subtype-specific findings. Therefore, we prioritized a comprehensive analysis encompassing the entire patient population to provide a more robust and generalizable overview of vaccination behaviors. We acknowledge the importance of subtype-specific data and agree it would enrich future studies with larger cohorts. We appreciate your understanding of these methodological considerations.
Comment 13: The authors should explore factors influencing actual vaccination uptake using multivariate analysis.
Response 13: We thank the reviewer for the thoughtful recommendation to explore factors influencing actual vaccination uptake using multivariate analysis. We fully agree that such an approach could offer valuable insights into determinants of vaccine behavior. However, as noted in our study, the overall vaccination rate among respondents was exceedingly low, resulting in a pronounced imbalance between vaccinated and unvaccinated individuals. This disparity poses significant limitations for conducting robust multivariate analysis. Specifically, the small number of vaccinated participants undermines statistical power and increases the risk of unstable estimates, while the large proportion of unvaccinated individuals may introduce analytical bias. Given these constraints, we concluded that proceeding with multivariate modeling under such conditions could compromise the reliability of the findings and potentially lead to misinterpretation. In light of this, we have opted to maintain a descriptive and exploratory analytical framework to preserve the integrity of our conclusions. We appreciate your understanding of the methodological limitations inherent to our dataset and remain committed to pursuing more detailed multivariate analyses in future studies with larger, more balanced samples.
Comment 14: The role of physician recommendation in influencing vaccine uptake is notably absent and should be discussed.
Response 14: Thanks for your kind suggestion. We have expanded the Discussion section to address the critical role of physician recommendation in influencing vaccine uptake. Citing relevant literature, we emphasize that physician endorsement is a key determinant of vaccination behavior, particularly in adult and high-risk populations. We also note that proactive communication by healthcare providers during routine clinical encounters can significantly improve patients’ confidence in vaccine safety and efficacy. This addition underscores the importance of integrating vaccine education into standard rheumatologic care (Line 412-421). Relevant references have been cited to substantiate these points (PMID:39917737).
Comment 15: The discussion should identify potential weaknesses, the lack of information on provider behavior or healthcare access.
Response 15: Thanks for your kind suggestion. We have added a relevant discussion on the role of physician recommendation in influencing vaccine uptake in the Discussion section (Line 412-421, 432-437). In future research, we plan to conduct a survey to investigate the impact of healthcare provider behaviors on herpes zoster vaccination uptake among patients with rheumatoid diseases.
Comment 16: The discussion should include more specific suggestions for public health interventions, e.g., integrating vaccine education into rheumatology clinic visits or partnering with community pharmacies.
Response 16: Thank you for your kind suggestions for revision. We have expanded the Discussion section to incorporate targeted recommendations aimed at improving vaccine uptake. These include integrating vaccine education into routine rheumatology clinic visits to capitalize on patient-provider interactions, as well as establishing partnerships with community pharmacies to increase vaccine accessibility and awareness. Additionally, we emphasize the importance of enhancing patient knowledge through tailored educational materials and strengthening healthcare service capacity to support vaccination delivery (Line 424-437).
Comment 17: Bibliographic references should include DOI when available.
Response 17: Thank you for your suggestion. We have reviewed the reference list and added DOIs for all citations where available to enhance the completeness and accessibility of the bibliographic information.
We sincerely thank you for your consideration and support. We believe that the revisions have significantly improved the manuscript and addressed the reviewers’ concerns. We look forward to your favorable response.
Round 2
Reviewer 1 Report
Comments and Suggestions for Authors
Authors properly addressed all my previous comments. Now the paper is acceptable for publication.
Only a minor observations:
About hospitalizations, authors should consider also oncological patients. Among complications, they should also report Nervous System HZ, Ophthalmic HZ and HZ Otitis Externa, citing proper literature. Among discussions, authors should consider the possible impact of vaccination in decreasing hospital admissions, as reported in several paper across Europe and US recently published.
Author Response
Comment 1: About hospitalizations, authors should consider also oncological patients. Among complications, they should also report Nervous System HZ, Ophthalmic HZ and HZ Otitis Externa, citing proper literature. Among discussions, authors should consider the possible impact of vaccination in decreasing hospital admissions, as reported in several paper across Europe and US recently published.
Response 1: We thank the reviewer for these insightful and constructive suggestions. In response, we have made the following revisions to strengthen the manuscript:
Hospitalizations: We have expanded our discussion of herpes zoster–related hospitalizations to specifically include oncological patients, recognizing their heightened vulnerability due to immunosuppressive treatments and underlying malignancies (Line 90-94).
Complications: We have added a detailed description of key herpes zoster complications, including herpes zoster involving the nervous system, ophthalmic herpes zoster, and herpes zoster otitis externa. These additions are supported by citations from relevant and up-to-date literature (Line 65-72).
Impact of Vaccination: We have incorporated a discussion on the potential role of herpes zoster vaccination in reducing hospital admissions. This section references recent studies conducted in Europe and the United States that demonstrate a decline in HZ-related hospitalizations following the implementation of vaccination programs.
We are grateful for your recommendations and believe these revisions have meaningfully enhanced the depth and relevance of our manuscript (Line 374-380).
We sincerely thank you for your consideration and support. We believe that the revisions have significantly improved the manuscript and addressed the reviewers’ concerns. We look forward to your favorable response.
Reviewer 2 Report
Comments and Suggestions for Authors
I thank the authors for sharing the revised version of the manuscript. I appreciate the thoughtful integration of the comments provided. The only additional recommendation I would make is to consider introducing the concept of vaccine literacy within the discussion section, as previously suggested. This would strengthen the interpretation of the findings and their implications for public health strategies.
Author Response
Comment 1: The only additional recommendation I would make is to consider introducing the concept of vaccine literacy within the discussion section, as previously suggested. This would strengthen the interpretation of the findings and their implications for public health strategies.
Response 1: We sincerely thank the reviewer for this valuable recommendation. In response, we have incorporated the concept of “vaccine literacy” into the revised Discussion section to strengthen the interpretation of our findings and to highlight their relevance for public health planning (Line 384-394). We acknowledge that vaccine literacy is a critical determinant of informed decision-making and vaccine acceptance. Our survey assessed participants’ knowledge of herpes zoster and its vaccine through structured questions addressing fundamental aspects such as disease etiology, transmission, clinical manifestations, complications, risk factors, and vaccination. By contextualizing our findings within the framework of vaccine literacy, we aim to better inform strategies for education, communication, and intervention targeted at improving vaccination coverage, particularly among high-risk groups.
We sincerely thank you for your consideration and support. We believe that the revisions have significantly improved the manuscript and addressed the reviewers’ concerns. We look forward to your favorable response.
Reviewer 3 Report
Comments and Suggestions for Authors
Dear authors please incorporate your answer to these questions , into the discussion, that will improved the manuscript. You could write for example "In our study cohort, the three most prevalent rheumatic diseases were systemic lupus erythematosus 336 (32.4%), rheumatoid arthritis 272 (26.3%), and Sjögren’s syndrome 102 (9.9%). However, given the extensive range of rheumatic diseases represented and the relatively limited sample size for many subtypes, conducting detailed subgroup analyses presents considerable challenges. The small numbers within less common disease categories limit statistical power and reduce the reliability of subtype-specific findings. Therefore, we prioritized a comprehensive analysis encompassing the entire patient population to provide a more robust and generalizable overview of vaccination behaviors. We acknowledge the importance of subtype-specific data and agree it would enrich future studies with larger cohort
Comment 12: Authors should include specific analysis, such as by specific rheumatic disease subtypes.
Response 12: We sincerely appreciate the reviewer’s insightful feedback regarding the inclusion of analyses by specific rheumatic disease subtypes. In our study cohort, the three most prevalent rheumatic diseases were systemic lupus erythematosus 336 (32.4%), rheumatoid arthritis 272 (26.3%), and Sjögren’s syndrome 102 (9.9%). However, given the extensive range of rheumatic diseases represented and the relatively limited sample size for many subtypes, conducting detailed subgroup analyses presents considerable challenges. The small numbers within less common disease categories limit statistical power and reduce the reliability of subtype-specific findings. Therefore, we prioritized a comprehensive analysis encompassing the entire patient population to provide a more robust and generalizable overview of vaccination behaviors. We acknowledge the importance of subtype-specific data and agree it would enrich future studies with larger cohorts. We appreciate your understanding of these methodological considerations.
Comment 13: The authors should explore factors influencing actual vaccination uptake using multivariate analysis.
Response 13: We thank the reviewer for the thoughtful recommendation to explore factors influencing actual vaccination uptake using multivariate analysis. We fully agree that such an approach could offer valuable insights into determinants of vaccine behavior. However, as noted in our study, the overall vaccination rate among respondents was exceedingly low, resulting in a pronounced imbalance between vaccinated and unvaccinated individuals. This disparity poses significant limitations for conducting robust multivariate analysis. Specifically, the small number of vaccinated participants undermines statistical power and increases the risk of unstable estimates, while the large proportion of unvaccinated individuals may introduce analytical bias. Given these constraints, we concluded that proceeding with multivariate modeling under such conditions could compromise the reliability of the findings and potentially lead to misinterpretation. In light of this, we have opted to maintain a descriptive and exploratory analytical framework to preserve the integrity of our conclusions. We appreciate your understanding of the methodological limitations inherent to our dataset and remain committed to pursuing more detailed multivariate analyses in future studies with larger, more balanced samples.
Author Response
Comment 1: The authors should include specific analysis by rheumatic disease subtypes.
Response 1: We appreciate this valuable suggestion. In response, we have expanded the Discussion section to incorporate detailed analysis and interpretation related to specific rheumatic disease subtypes. This addition enhances the contextualization of our findings and addresses the heterogeneity among patient groups (Line 481-489).
Comment 2: The authors should explore factors influencing actual vaccination uptake using multivariate analysis.
Response 2: Thank you for this important recommendation. We have now added a detailed discussion in the Discussion section addressing the points you raised . We appreciate your understanding of these comments (Line 489-496).
We sincerely thank you for your consideration and support. We believe that the revisions have significantly improved the manuscript and addressed the reviewers’ concerns. We look forward to your favorable response.